# GSK-3 and Tau: A Key Duet in Alzheimer’s Disease

**DOI:** 10.3390/cells10040721

**Published:** 2021-03-24

**Authors:** Carmen Laura Sayas, Jesús Ávila

**Affiliations:** 1Instituto de Tecnologías Biomédicas (ITB), Universidad de La Laguna (ULL), 38200 Tenerife, Spain; 2Centro de Biología Molecular Severo Ochoa (CBMSO), Consejo Superior de Investigaciones Científicas (CSIC) y la Universidad Autónoma de Madrid (UAM), 28049 Madrid, Spain; 3Centro de Investigación Biomédica en Red de Enfermedades Neurodegenerativas (CIBERNED), Valderrebollo 5, 28031 Madrid, Spain

**Keywords:** GSK-3, tau phosphorylation, neurodegeneration, Alzheimer’s disease

## Abstract

Glycogen synthase kinase-3 (GSK-3) is a ubiquitously expressed serine/threonine kinase with a plethora of substrates. As a modulator of several cellular processes, GSK-3 has a central position in cell metabolism and signaling, with important roles both in physiological and pathological conditions. GSK-3 has been associated with a number of human disorders, such as neurodegenerative diseases including Alzheimer’s disease (AD). GSK-3 contributes to the hyperphosphorylation of tau protein, the main component of neurofibrillary tangles (NFTs), one of the hallmarks of AD. GSK-3 is further involved in the regulation of different neuronal processes that are dysregulated during AD pathogenesis, such as the generation of amyloid-β (Aβ) peptide or Aβ-induced cell death, axonal transport, cholinergic function, and adult neurogenesis or synaptic function. In this review, we will summarize recent data about GSK-3 involvement in these processes contributing to AD pathology, mostly focusing on the crucial interplay between GSK-3 and tau protein. We further discuss the current development of potential AD therapies targeting GSK-3 or GSK-3-phosphorylated tau.

## 1. Introduction

Glycogen synthase kinase-3 (GSK-3) is a ubiquitous serine/threonine kinase initially shown to phosphorylate and inhibit glycogen synthase [1]. GSK-3 has many substrates, including metabolic proteins [2], transcription factors [3], translation factors [4] or cytoskeletal proteins [5,6,7] present in different subcellular compartments.

GSK-3 is evolutionarily conserved in all eucaryotes examined to date [8,9,10]. There are two highly similar GSK-3 isoforms in mammals, GSK-3α and GSK-3β [11,12], which are known to act on different substrates [13]. GSK-3β is enriched in the brain and shows increased levels during aging [14]. Moreover, alternative splicing of the GSK3β gene in rodents [15] and humans [16,17,18] leads to a long form named GSK3β2. GSK-3β2 is neuron-specific, with high expression during brain development that persists until adulthood [19,20]. 

GSK-3 has been traditionally considered a constitutively active kinase, with high activity in resting cells, and inhibited by serine phosphorylation (Ser 21 in GSK-3α and Ser 9 in GSK-3β), induced by numerous extracellular signals and mediated by different ser/thr kinases [2,21,22,23]. However, GSK-3 activation has also been shown to occur in association with an increase in tyrosine phosphorylation (Tyr 279 in GSK-3α and Tyr 216 in GSK-3β). GSK-3 tyrosine phosphorylation has been proposed to occur either by intramolecular autophosphorylation [24,25], or by the action of tyrosine kinases such as Fyn [26] or Pyk2 [27,28] in mammals, or ZAK1 in *D. discoideum* [10]. GSK-3 can also be activated by serine dephosphorylation by phosphatases such as PP2A [29]. GSK-3 activity can be regulated by other mechanisms including protein complex association (e.g., in the Wnt or the Hedgehog pathways [30,31]), subcellular localization [32,33] and protein cleavage [34,35]. Of note, in neuronal cells, GSK-3 has been shown to be activated by extracellular factors that induce growth cone collapse and neurite retraction, such as lysophosphatidic acid (LPA) [28,36] or semaphorin 3A [37] and by neurotoxic insults such as prion protein (PrP) [38] or amyloid-β (Aβ) peptide [39].

Diverse signaling pathways converge on GSK-3 during neuronal development. For this reason, GSK-3 is a key regulator of several processes involved in the development of neurons, such as neurogenesis [40], regulation of neuronal polarity and axon extension [41,42,43,44], dendrite development [45], synaptogenesis [46], and neuronal survival. Furthermore, studies performed in GSK-3 mutant mice have shown the importance of GSK-3 in different aspects of brain function, including brain development, learning and memory, emotionality, depressive-like behavior, and sensorimotor functions (reviewed in [47]). GSK-3β knockout (KO) mice die during late embryonic development due to liver and heart defects [48], but the brain and spinal cord do not show gross abnormalities [49]. However, GSK-3β heterozygous mice present diverse behavioral deficits, such as reduced movement, increased anxiety, aggressive behavior, and decreased memory [50,51,52]. Downregulation of GSK-3β by shRNA injection in the hippocampal dentate gyrus results in decreased immobility time and anti-depressant-like behavior [53]. Behavior is also altered in transgenic mice that overexpress GSK-3β in the brain, which show mania and hyperactivity [54]. Of note, GSK-3α KO animals are viable, but male mice are infertile [55,56]. These mice also show behavioral deficits such as decreased immobility time, decreased exploratory activity and anti-aggression behavior or impaired associative memory and coordination, among others [57]. Although research in the field has mostly focused on the GSK-3β isoform, recent evidence points to GSK-3α as a key player in important brain processes such as synaptic plasticity [58].

## 2. Interplay between GSK-3 and Tau in Alzheimer’s Disease

Since GSK-3 plays central roles in nervous system function, GSK-3 dysregulation is involved in the pathogenesis of diverse brain disorders, including Alzheimer’s disease (AD). AD is the most prevalent neurodegenerative disorder, characterized by a progressive loss of episodic memory as well as behavioral and cognitive deficits. Some GSK-3 targets are of particular relevance to AD, including Aβ peptide and tau protein, the components of the main pathological hallmarks of AD, namely neuritic plaques (NPs) and neurofibrillary tangles (NFTs), respectively [5,59,60]

GSK-3 has been proposed to function as a molecular link between Aβ and tau in AD pathogenesis [61]. Aβ activates GSK-3 [62,63], which in turns phosphorylates tau [64]. On the other hand, GSK-3 regulates amyloid precursor protein (APP) metabolism and Aβ production and promotes neuronal death induced by Aβ [65,66,67]. In diverse mouse models of AD, GSK-3 inhibition with different compounds reduces Aβ deposition and neuritic plaque formation, as well as tau phosphorylation, and improves cognitive deficits, as measured in behavioral assays [68,69,70,71,72]. Conversely, mice conditionally overexpressing GSK-3β in the forebrain (Tet/GSK-3β mice) show hyperphosphorylation of tau in AD relevant epitopes, in correlation with somatodendritic accumulation of tau in hippocampal neurons [73]. Tet/GSK-3β mice also display different signs of neurodegeneration such as increased neuronal cell death, reactive astrocytosis, and microgliosis [73], as well as spatial learning defects [74]. Transgenic mice conditionally overexpressing GSK-3β recapitulate different aspects of AD pathology, thus providing a suitable animal model to study the relevance of GSK-3β dysregulation for AD pathogenesis. Shutdown of the transgene in Tet/GSK-3β mice with signs of AD pathology leads to normal GSK-3 activity, normal levels of phosphorylated tau, reduced neuronal death and reactive gliosis, and improvement of cognitive deficits [75]. Remarkably, mice overexpressing GSK-3β in the absence of tau (tau knockout background) present milder and slower signs of neurodegeneration as well as reduced cognitive deficits [76]. Recent data indicate that transgenic mice with GSK-3β haploinsufficiency show reduced tau hyperphosphorylation, synaptic accumulation, aggregation, and trans-cellular spread in the brain [77]. This suggests that the AD-like pathology found in brains of GSK-3 overexpressing mice relies, at least partially, on hyperphosphorylated tau (see Figure 1). 

Elevated levels of GSK-3 have been found in postmortem brains from AD patients compared to age-matched control samples [78]. Abnormal activation of GSK-3, for example, by truncation, has also been associated with AD [79]. Of note, LPA, a reported activator of GSK-3 and tau phosphorylation [28,36], has been related to AD. Different LPA species showed significant positive association with cerebrospinal fluid (CSF) biomarkers of AD, such as Aβ-42, phospho-tau, and total tau [80]. In addition, autotaxin, the enzyme that catalyzes the formation of LPA, shows increased levels in CSF and brains from AD patients [81,82]. This suggests that dysregulation of LPA metabolism might be linked to GSK-3 overactivation and tau phosphorylation in AD brains.

## 3. Tau Protein: Functional Domains, Isoforms, and Phosphorylation

Tau is a microtubule (MT)-associated protein [83,84], mainly expressed in neurons [85]. Tau binds along the MT lattice and regulates MT assembly, stability, and dynamics in neurons (reviewed in [86]). Recent work suggests that tau preferentially binds GDP-tubulin at the MT polymer [87] and that tau enables axonal MTs to have long labile domains [88,89]. Tau protein shows different functional domains [90]. The longest tau isoform (tau 2N4R, 441aa, Figure 2) contains an N-terminal projection domain with two inserts (N1 and N2), a proline-rich region (PRR), a MT binding region (MTBR), and a C-terminal domain. The MTBR is formed by three or four imperfect repeat sequences: R1, R2, R3, and R4 (Figure 2) [91].

Tau protein is encoded by a single-copy gene, MAPT, located in chromosome 17 in human, and chromosome 11 in mice. MAPT undergoes alternative splicing of exons 2,3, and 10, giving rise to six different tau isoforms [91,92]. Exon 10 codes for one of the four repeats that form the microtubule-binding region (MTBR) (Figure 2). Tau isoforms differ in their number of MTBR repeats, containing three (tau 3R) or four (tau 4R), and in the number of N-terminal inserts (zero, one or two), localized in the N-terminal region [93]. The expression of the different tau isoforms is developmentally regulated [94]. Differences are found in the expression patterns of human and murine isoforms. Tau 3R is the predominant isoform in the fetal brain, both in mice and humans [95]. However, while in the adult mouse brain, only tau4R is expressed, in the human brain, the expression of tau4R increases to the level of tau3R during the postnatal period, reaching a 1:1 ratio [92]. Deviations of this ratio are linked to diverse neurodegenerative diseases [96]. Human and murine tau also show some variability in their primary sequences, mostly in their N-terminal ends [97]. The N-terminal projection region of tau protrudes from the microtubule surface [98], allowing its interaction with membranes and with other proteins [99]. The N-terminal region of human tau contains an additional 11aa peptide, which is absent in mouse tau (aa17-28) and enables human tau to interact with specific protein partners [100,101]. Tau protein has been proposed to adopt a “paperclip” conformation in solution in which the N- and the C-terminal domains fold back on the central and MTBR domain [102]. The longer N-terminus might influence the intramolecular interaction between the N- and C-terminal ends of the protein and the MTBRs, making human tau more predisposed than mouse tau to adopt this pathological “paperclip” conformation. This would contribute to explaining why humans are particularly susceptible to develop tau pathology (including NFTs), leading to neurodegeneration. It further explains why mouse models fail to recapitulate tau pathology, unless human tau is overexpressed.

Tau binding to MTs is mostly regulated by phosphorylation [103,104,105]. Tau is a substrate of several kinases, mostly ser/thr kinases (such as GSK-3, CDK5, MARK, PKA, CamKII, PKC, MAPK, JNK, or ROCK), but also tyrosine kinases (Fyn or Abl), that phosphorylate tau at numerous specific sites (~85) (reviewed in [106]. Figure 2 shows schematically the best characterized tau phosphorylation sites, highlighting GSK-3 sites (more than 30). Tau phosphorylation reduces its affinity for MTs, providing dynamics to the system in healthy neurons [103,104]. In AD brains, tau is hyperphosphorylated on serines and threonines in paired helical filaments (PHFs) in NFTs [107]. GSK-3 phosphorylates most residues present in hyperphosphorylated tau in AD (reviewed in [108]). Tau binds to both GSK-3 isoforms, GSK-3α and GSK-3β, in brain extracts, although the complex tau/GSK-3β is more abundant [109]. The tau–GSK-3 interaction may be direct [109] or mediated by 14-3-3 protein [110]. Tau interaction with 14-3-3 protein is enhanced upon tau phosphorylation [111]. Tau phosphorylation by GSK-3 lies at the core of diverse neuronal processes that are dysregulated during AD pathogenesis.

## 4. Axonal Transport Impairment by GSK-3-Mediated Tau Phosphorylation

Axonal transport disruption is an early pathological hallmark common to many neurodegenerative disorders. Disturbance of axonal trafficking leads to subcellular mislocalization and aggregation of molecules, organelles, and synaptic vesicle in affected neurons, contributing to synaptic failure, neuronal malfunction, and eventual degeneration. Tau axonal transport relies on its interaction with kinesin-1, which is dependent on GSK-3 phosphorylation [112]. Of note, axonal transport is disrupted upon tau overexpression, leading to vesicular aggregation, a process reversed by GSK-3 inhibition [13]. Furthermore, GSK-3 regulates axonal trafficking of mitochondria in a tau-dependent manner [113]. 

Efficient transport in axons largely depends on an intact MT cytoskeleton. Hyperphosphorylated tau has been found to mediate axonal transport defects under pathological conditions such as AD [114]. Tau hyperphosphorylation induces MT disassembly, which impairs motor protein and cargo binding [115]. A proposed mechanism for this phenomenon is that tau hyperphosphorylation leads to impairment of the function of c-Jun N-terminal kinase-interacting protein 1 (JIP1), a protein that regulates cargo binding to kinesin motors. As observed in AD brains, hyperphosphorylated tau interacts with JIP1, blocking JIP1 axonal transport and inducing its accumulation in the cell body [114]. 

## 5. GSK-3 Regulates the Cholinergic Function: Involvement of Tau Phosphorylation

AD is characterized by a progressive cognitive decline, which is in part due to an impairment in cholinergic neurotransmission. AD brains show a selective loss of cholinergic neurons in specific brain regions and a reduction in acetylcholine (ACh) levels [116]. Moreover, the activity of choline acetyltransferase (ChAT), the enzyme that mediates ACh synthesis, is reduced in brains of AD patients in correlation with the severity of the dementia [117]. GSK-3 has been shown to be a regulator of the cholinergic function. GSK-3 activation induces a reduction in ACh levels in striatum [118], nucleus basalis of Meynert and frontal cortex [119]. The GSK-3-induced reduction in ACh levels is mediated by inactivation of ChAT [118,119]. GSK-3 phosphorylates and inactivates pyruvate dehydrogenase (PDH), an enzyme that catalyzes the conversion of pyruvate to acetyl CoA in mitochondria, thus leading to a reduction in Ach in cholinergic neurons [120]. Notably, GSK-3 disturbs axonal transport in cholinergic neurons, in correlation with tau hyperphosphorylation, and the subsequent accumulation of ChAT in cell bodies [119]. Cholinergic immunotoxin 192-IgG saporin induces degeneration of basal forebrain cholinergic neurons, leading to an increase in GSK-3 activation and the subsequent tau phosphorylation [121]. 

## 6. GSK-3 and Tau Roles in Adult Hippocampal Neurogenesis

The hippocampus is the brain area that shows the highest degree of plasticity during adulthood, providing the substrate for hippocampal-dependent learning and memory acquisition and maintenance [122,123]. The reason for this vast plasticity is the continuous addition of new neurons to the hippocampus throughout life in a process known as adult hippocampal neurogenesis (AHN) [124,125]. Recent evidence indicates that AHN persists during both physiological and pathological aging in humans [126]. However, the number and maturation of new neurons gradually decay with the advance of the AD pathology [126]. Therefore, impaired AHN in the hippocampus might contribute to memory deficit and cognitive decline in AD. 

GSK-3β has emerged as one of the key regulators of AHN. Voluntary exercise in mice has been shown to induce an increase in AHN and improved cognition in correlation with activation of GSK-3β [127]. Other reports indicate that inhibition of GSK-3β promote AHN in vitro and in vivo [128,129]. Cerebral ischemia-induced AHN involves GSK-3β inactivation downstream of the PI3-K/Akt pathway [130]. Dysregulated hyperactive GSK-3 impairs in vivo neurogenesis in mice and the capacity of therapeutic agents to stimulate neurogenesis [131]. Of note, conditional overexpression of GSK-3β in mice (Tet/GSK3β mice) leads to a number of defects in AHN, such as non-reversible depletion in neurogenic niches in the dentate gyrus of hippocampus, impairment in the differentiation and maturation of newborn neurons, delay in the switching-off of doublecortin expression, as well as alteration in the survival and death rates of immature precursors [132]. Interestingly, GSK-3β overexpression in mice (Tet/GSK3β mice) causes reversible alterations to the dendritic morphology and postsynaptic densities of hippocampal granule neurons, similar to the morphological defects found in hippocampal neurons of AD patients [133] (see Figure 3). In line with this, GSK-3 inhibitors have been shown to improve the cognitive function in mouse models of different diseases such as AD, Down syndrome (DS), or Fragile X syndrome, by partially repairing defects in neurogenesis that occur in these disorders [134]. Tau has been shown to play key in vivo roles in the morphological and synaptic maturation of newborn hippocampal granule neurons in mice [135]. 

Tau deficiency in mice leads to impairment in the maturation of newborn granule neurons under basal conditions and these neurons are unresponsive to stimuli that modulate AHN, such as environmental enrichment [135]. Tau also underlies the effects of stress on AHN in mouse brain [135,136]. During the pathogenesis of AD and other tau-related dementia, changes in soluble tau species, including tau phosphorylation, lead to neuronal death [137]. Notably, soluble tau, mostly composed of monomers and dimers, induces long-term detrimental effects on the morphology and connectivity of newborn granule neurons, in correlation with impaired behavioral pattern separation skills [138]. Overall, these data suggest that at least part of the observed effects of GSK-3β in AHN might be mediated by its target protein tau (see Figure 3).

## 7. GSK-3 Modulation of Cognitive Functions

Cognitive impairment and memory loss in AD correlate with synaptic dysfunction (reviewed in [139]) and eventual synaptic loss (reviewed in [140]). 

GSK-3 has been implicated in the modulation of cognitive functions both at the presynaptic and postsynaptic levels. At the presynaptic level, GSK-3 upregulation markedly reduces the presynaptic release of glutamate and the expression/clustering of synapsin I, a synaptic vesicle protein involved in neurotransmitter release [141]. GSK-3 inhibits exocytosis of synaptic vesicles by phosphorylating P/Q type calcium channels and impairing the formation of SNARE complexes [142]. Additionally, GSK-3 controls activity-dependent bulk endocytosis of synaptic vesicles, through phosphorylation of the large GTPase dynamin [143].

Additionally, GSK-3 regulates postsynaptic function by modulating synaptic plasticity [144]. Synaptic plasticity is the activity-dependent modification of the transmission efficiency of the synapses (strengthening or weakening), which provides an essential component for learning and memory [145]. The structural basis of synaptic plasticity in the adult brain relies on the formation of new dendritic spines as well as the dynamic changes in spine morphology. Long-lasting potentiation of synaptic strength by repetitive activation of excitatory synapses is termed long-term potentiation (LTP). LTP is triggered by intense activation of N-methyl-D-aspartate glutamate receptors (NMDARs) that causes the incorporation of α-amino-3-hydroxy-5-methylisoxazole-4-propionic acid glutamate receptors (AMPARs) into the postsynaptic membrane. Long-term depression (LTD) is a long-lasting decrease in the efficiency of synaptic transmission [146,147], by weak activation of NMDRs, resulting in the removal of AMPARs from the postsynaptic membranes. This leads to a decrease in synaptic efficiency which eventually can lead to synapse shrinkage and elimination [147]. The prototypic forms of synaptic plasticity are the LTP and LTD that occur in the CA1 region of the hippocampus. LTP and LTD are considered the molecular and cellular basis of learning and memory processes. Most excitatory synapses localize on dendritic spines, and undergo growth in response to LTP and elimination following LTD.

GSK-3 participates in the regulation of synaptic plasticity by modulating LTP and LTD (reviewed in [144,148]). In this way, GSK-3 regulates cognitive function directly. GSK-3β is inhibited upon LTP induction, as denoted by an increase in Ser 9 phosphorylation [149,150]. Of note, transgenic mice expressing the phosphorylation defective mutant GSK3ss[S9A] (constitutively active GSK-3β) show a marked inhibition of LTP, in correlation with learning deficits [74]. Transgenic mice that overexpress GSK-3β also present impaired LTP, which is reversed by GSK-3 inhibition [149]. In addition, activation of GSK-3 inhibits LTP with synapse-associated impairments [141].

On the other hand, GSK-3 is involved in the induction of NMDAR-mediated LTD [150]. GSK-3 inhibitors prevent LTD induction in rat hippocampal slices [150]. GSK-3 is activated during LTD, most likely due to protein phosphatase 1 direct dephosphorylation and by Akt inhibition, downstream of calcium entry through NMDARs [150].

NMDARs are cation channels gated by the neurotransmitter glutamate, involved in synaptic formation, synaptic plasticity, learning and memory as well as in excitotoxicity (reviewed in [151]). GSK-3 interplays with NMDARs in different ways. GSK-3 may be a NMDAR target since stimulation of NMDARs induces calpain-mediated cleavage and activation of GSK-3 [34]. Conversely, NMDARs have been reported to be a target of GSK-3, as NMDAR currents are reduced by a variety of GSK-3 inhibitors and upon GSK-3 knockdown [152]. Moreover, GSK-3 regulates NMDAR internalization and function, mostly the receptors containing NR2B subunits [152]. Overactivation of GSK-3β induces a reduction in the expression of NMDAR subunits NR2A/B and the scaffolding protein Postsynaptic density-93 (PSD-93) at synapses [141]. GSK-3 also influences the interaction between NMDARs and PSD-95 [152]. Notably, although the roles of GSK-3 in the brain have traditionally been assigned to GSK-3β, it has recently been reported that GSK-3α, not GSK-3β, is the isoform required for NMDAR-dependent regulation of LTD [58].

GSK-3 has been shown to participate in the crosstalk between LTP and LTD [148]. GSK-3 inhibitors rescue abnormal LTP and/or LTD in neuropsychiatric disorders, thereby improving cognitive deficits [134]. Thus, inhibition of GSK-3 may improve cognitive dysfunction in some conditions by regulating hippocampal synaptic plasticity.

Besides its role as a regulator of NMDARs, GSK-3 is also implicated in the regulation of AMPARs. AMPARs mediate fast excitatory neurotransmission and play a key role in synaptic plasticity, being involved in LTP and LTD of hippocampal synaptic transmission (reviewed in [153]). GSK-3 inhibition or knockdown reduces AMPAR synaptic responses, in correlation with a loss of AMPAR surface localization and an increase in AMPAR internalization [154]. Modulation of AMPAR internalization by GSK-3 is mediated by the GDI-Rab5 complex [154]. GSK-3 regulation of AMPAR trafficking and function provides an extra layer of regulation of synaptic transmission and plasticity by GSK-3. Overall, these studies indicate that GSK-3 exerts critical functions in synaptic assembly and function.

## 8. Interplay between GSK-3 and Tau during LTD

Interplay between GSK-3 and tau during LTD diverse GSK-3 substrates participate in GSK-3 mediated LTD. Phosphorylation of phosphatidylinositol 4 kinase type IIa (PI4KIIa) by GSK-3 is involved in stabilization of NMDARs at synapses, regulation of cell-surface expression of AMPARs, and direction of vesicular trafficking to lysosomes [155,156]. PSD-95 is phosphorylated by GSK-3 and undergoes synaptic detachment, a process necessary for LTD induction [157]. Another GSK-3 substrate involved in LTD is protein interacting with C kinase 1 (PICK1), a protein that interacts with AMPARs [158]. GSK-3-phosphorylated PICK1 enhances its interaction with AMPARs, this contributing to retain AMPARs internalized during LTD [159,160]. Kinesin light chain 2 (KLC2) is also phosphorylated by GSK-3 and participates in AMPARs intracellular trafficking [161]. The signaling pathway involving GSK-3/KLC2/AMPARs may contribute to the regulation of synaptic plasticity downstream of diverse neurotransmitters and growth factors during learning and memory and in the pathophysiology of some psychiatric conditions [161,162,163,164].

Tau protein has emerged as a potential key candidate for the multifaceted actions of GSK-3 in LTD. Of note, tau has been found to be required for LTD induction in the hippocampus [165], and impaired LTD has been reported in a tauopathy mouse model (THY-tau22 mice) [166]. Tau is involved in LTD induced by physiological stimuli such as insulin [167] or glucocorticoids (GCs) [168]. Tau deletion leads also to severe deficits in LTP [169]. Most NMDARs localize at glutamatergic synapses, where they exert their actions as mediators of synaptic transmission and synaptic plasticity. Remarkably, a proportion of NMDARs are present outside synapses in the plasma membrane. Although extrasynaptic NMDARs have been mostly linked to excitotoxicity and cell death [170], extrasynaptic NMDAR currents are also crucial for efficient LTD [171,172,173]. Of note, tau knockout neurons lack NMDA extrasynaptic currents [174]. This may contribute to LTD deficits exhibited in tau knockout mice, which also show spatial reversal learning [175]. Furthermore, tau participates in the regulation of the interaction between PICK1 and AMPARs, thereby contributing to other key processes for LTD, such as AMPARs internalization [175]. 

Remarkably, GSK-3α specific implication in LTD occurs via its transient anchoring in dendritic spines in a tau-dependent manner [58]. GSK-3α overexpression leads to LTD only in the presence of tau, suggesting that tau is downstream of GSK-3α in this process [58]. GSK-3 phosphorylates tau during LTD [165], and phosphorylation of tau in a GSK-3 site (Ser 396) is essential for LTD [175]. Therefore, GSK-3-mediated tau phosphorylation and GSK-3α accumulation in dendritic spines may be crucial events to trigger LTD. Tau actions on AMPAR internalization and on NMDA extrasynaptic currents contribute further to LTD, as mentioned above. The interplay between GSK-3α and tau during NMDAR-mediated LTD is complex and requires further investigation (see Figure 4).

Of note, tau interacts with and is phosphorylated also by tyrosine kinases from the Src family, such as Fyn [176], which is also present in NFTs in AD brains [177]. Tau is necessary for the transport of Fyn to dendritic spines, where Fyn phosphorylates the GluN2B subunit of NMDAR [178]. Fyn phosphorylation enhances the interaction between NMDAR and PSD-95 in synapses. Stabilization of the NMDAR/PSD95 complex underlies Aβ-induced excitotoxicity in AD animal models [178]. Tau contributes to the synaptic localization of kinases to trigger different signaling pathways involved in synaptic plasticity and/or excitotoxicity. 

Tau is predominantly localized in axons under physiological conditions, where it regulates MT stability and promotes MT polymerization [94]. Therefore, tau requirement for physiological LTD is surprising since LTD is a postsynaptic process, mediated by the synaptic removal of AMPARs. One possibility is that LTD might induce tau relocalization to dendritic shafts and/or spines. Another option, supported by recent evidence, is that some tau is physiologically present in dendritic spines [174], and that this proportion of tau is the one that mediates LTD, exerting its function in synaptic plasticity. In addition, mechanical injuries of neurons lead to mislocalization of tau to dendritic spines and tau-dependent synaptic dysfunction, mediated by GSK-3 (and CDK5) tau phosphorylation [179]. In neurodegenerative disorders such as AD and other dementias, tau becomes hyperphosphorylated, detaches from MTs. and is missorted from axons to the somatodendritic compartment, where it interferes with synaptic function [180,181].

## 9. GSK-3 and GSK-3-Phosphorylated Tau as Therapeutic Targets in AD

In the last decades, the development of potential therapies for AD has been centered in the counteraction of the formation of Aβ plaques, since Aβ was considered the key factor in the pathogenesis of the disease. However, the progression of the cognitive decline that occurs in AD has been shown to correlate much better with the propagation of tau pathology than with the deposition of Aβ plaques [182,183]. For that reason, the focus of AD drug discovery has been shifted toward tau-targeting therapies. 

Hyperphosphorylated tau is the main component of the toxic forms of tau present in AD, including aggregates, fibrils, or NFTs [184]. Moreover, the correct interplay between GSK-3 and tau is crucial for the proper functioning of diverse neuronal and brain processes, which, when dysregulated, participate in AD pathogenesis. Therefore, GSK-3 has become an attractive therapeutic target in AD. An intense research effort has been made in recent years to develop novel small molecules or peptides and to identify natural compounds that inhibit GSK-3 and that could be used as potential therapies for AD [185]. Inhibitors of GSK-3 activity lie in different categories including cations (e.g., lithium), ATP competitive inhibitors, non-ATP competitive inhibitors (such as substrate competitive inhibitors (SCIs) or allosteric inhibitors) (reviewed in [186]. Small molecule inhibitors have been mostly designed by computer-based docking approaches. As mentioned above, GSK-3α and GSK-3β have different roles in key neuronal processes such as LTD [58]. Since most known inhibitors do not show isoform specificity, finding small molecules that inhibit one of the GSK-3 isoforms specifically is needed. Moreover, as GSK-3 participates in diverse key cellular processes, such as cell division, apoptosis, metabolism, or differentiation, a complete inhibition of GSK-3 may be toxic and induce side effects such as hypoglycemia or tumor progression. For that reason, compounds that induce a partial inhibition of GSK-3 activity should be identified for use as potential therapies in AD (or other disorders involving GSK-3). Substrate competitive inhibitors (SCIs) have been proposed as more suitable candidates for therapies, as they show higher specificity than ATP-competitive inhibitors and weak inhibitory activity. Among the most promising SCIs candidates are the novel kinase-peptide inhibitor-based SCI molecules which have been designed based on a ligand-protein binding model [187].

Several GSK-3 inhibitors have been shown to reduce tau phosphorylation in cells and in preclinical studies in mice (reviewed in [188]). However, despite the high number of GSK-3 inhibitors developed over the last years, only a few have reached clinical trials in humans, and none of them have made it to the market. One of these inhibitors is tideglusib, an irreversible non-ATP competitive GSK-3 inhibitor that acts as an allosteric inhibitor [189]. Tideglusib showed neuroprotective effects by reducing amyloid deposition, gliosis, tau phosphorylation and neuron loss, and by reversing spatial memory deficits in transgenic mice in preclinical studies. Tideglusib reached phase II clinical trials, and was shown to be generally well tolerated and safe, with significant improvement (at a particular concentration) in cognitive function in patients with mild-to-moderate AD, previously treated for months with cholinesterase inhibitors [190]. Nevertheless, in general, tideglusib did not exhibit obvious therapeutic effects for AD in clinical trials. Drug discovery towards novel GSK-3 inhibitors that can be effectively used for treating AD remains one of the challenges in the field. 

The development of potential tau-targeting therapies for AD includes different types of molecules, which range from agents that modulate tau posttranslational modifications (such as phosphorylation, -including GSK-3 inhibitors-, acetylation, or O-GlcNAcylation); compounds that stabilize MTs; activators of tau degradation by autophagy; inhibitors of tau aggregation; and agents that decrease tau expression (antisense oligonucleotides (ASOs) (reviewed in [191,192]). However, since the discovery of the importance of transcellular tau spreading in the progression of tau pathology in AD, the focus has been placed on tau-based immunotherapy (both passive and active), an approach to target (mostly but not only) extracellular tau, thereby stopping the spreading process. Immunotherapy has been shown to decrease tau pathology and improve cognitive deficits in preclinical studies in AD mouse models (reviewed in [193]). 

Passive immunotherapy implies the administration of antibodies or antisera against a specific tau epitope, eliminating the need of the recipient to produce an immune response and generate his/her own antibodies. Some of the antibodies used in passive immunotherapy target phospho-tau—in particular, tau phosphorylated in GSK-3 sites (eg. epitope pSer396/pSer404 (PHF-1 antibody) [194,195]; epitope pThr212/pThr217; [196,197] (antibodies JNJ-63733657, and PT3 and its humanized version hTP3, [198]; pThr231 and pSer396 (antibody PHF-13) [199]. Targeting phospho-tau at some of these epitopes (e.g., pSer396) may be of special relevance to delay AD progression at initial phases since they are specifically hyperphosphorylated at earlier stages of the disease [200]. The potential therapeutic actions of these antibodies have been tested in vivo in preclinical mouse models showing to reduce tau pathology and functional deficits (reviewed in [201]). Of note, one of these antibodies JNJ-63733657 has reached the clinical phase. JNJ-63733657 is a monoclonal antibody that selectively recognizes PHF tau and the mid-region of tau with high affinity for pThr212 and pThr217. In the first phase I clinical trial (NCT03375697), JNJ-63733657 was shown to be generally safe and well-tolerated and to reduce cerebrospinal fluid (CSF) phospho-tau levels in a dose-dependent manner [202]. No results have been reported yet from the second phase I trial (NCT03689153), which ended in December 2019, performed in both healthy participants and participants with prodromal or mild AD. JNJ-63733657 is currently being tested in a phase 2 study (NCT04619420) that started in January 2021 in people with early AD symptoms and a positive tau PET scan. The ongoing trial will run until 2025 and besides safety and pharmacokinetics, tau pathology burden and measures of cognition will be assessed. 

Another promising tau-targeting therapy for AD is active immunization by the administration of tau-directed vaccines. These vaccines aim to elicit an immune response targeted to specific pathological conformers of tau or phosphorylated tau without inducing autoimmune responses against physiological forms of tau. One of the generated vaccines, ACI-35, is a liposome-based vaccine that encompasses 16 copies of a synthetic fragment of pSer396/pSer404 tau (residues phosphorylated by GSK-3), attached into a lipid bilayer [203]. ACI-35 elicits a rapid and robust immune response in preclinical studies in transgenic tau mice and a reduction in tau pathology [204]. Notably, ACI-35 was the first vaccine against phospho-tau tested in clinical trials. ACI-35 showed no safety concerns but elicited a weak response in people with mild-to-moderate AD, in a phase I clinical trial (ISRCTN13033912). These results prompted the generation of a redesigned version of the vaccine that induces an enhanced immune response, with the production of antibodies specific to phosphorylated tau, and recognition of PHFs extracted from AD brain. An intermediate report from a currently ongoing Phase1b/2a clinical trial for ACI-35.030 in people with early AD (NCT04445831) pointed to no safety problems and good immunogenicity with generation of anti phospho-tau antibodies. Further research efforts on tau-based immunotherapies are needed in the hope to obtain safe and efficient therapies for AD.

## 10. Conclusions and Future Perspectives

GSK-3 is a crucial regulator of several neuronal processes that are dysregulated in AD, including axonal transport, cholinergic function or synaptic plasticity. GSK-3 has numerous substrates in neurons. GSK-3 promiscuity makes it difficult to elucidate whether the observed effects of GSK-3 are directly due to its kinase activity or to indirect actions via GSK-3 downstream effectors. This may be of relevance to understand specific neuronal and brain functions of the kinase and to design GSK-3 based therapies for AD and other GSK-3-related disorders. Although GSK-3 has many targets, tau protein seems to be the protein substrate that underlies many of these GSK-3 actions. Thus, GSK-3 and tau stand as a key duet in AD pathogenesis. 

GSK-3 and tau proteins have emerged as important potential therapeutic targets in AD. Obtaining specific and weak inhibitors of GSK-3 activity remains a key issue in the field. In the case of tau protein, reducing tau levels has been shown to be neuroprotective. However, tau is a multifaceted protein with diverse functions in neurons [86,205], beyond its actions as a MT-associated protein [206]. Hence, tau-based therapies face the challenge of targeting only pathological tau (including hyperphosphorylated tau), without altering physiological tau. Otherwise, these treatments could be deleterious, as in the case of potent GSK-3 inhibitors. Thus, enormous drug-discovery efforts are currently ongoing to develop potentially safe and efficient therapies for AD that target these two key proteins, such as GSK-3 inhibitors [187,191] and different types of agents targeting phospho-tau protein, including passive and active immunotherapy [191,192]. The final goal is to prevent disease progression at early stages before the brain damage is irreversible. To achieve this, it is also crucial to develop in parallel novel reliable biomarkers for an early detection of the disease.

## Figures and Tables

**Figure 1 cells-10-00721-f001:**
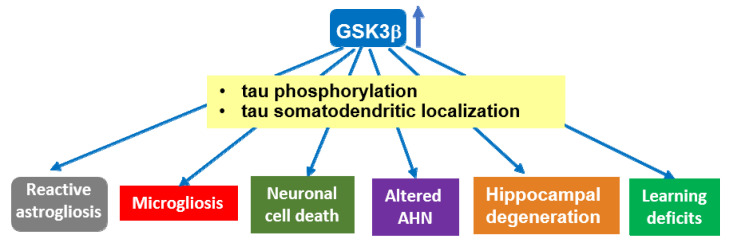
Overexpression of GSK-3β induces tau-dependent AD-like pathology. Transgenic mice overexpressing GSK-3β (Tet/GSK-3β mice) in forebrain show tau hyperphosphorylation and relocalization to the somatodendritic compartment in hippocampal neurons, in correlation with different signs of neurodegeneration. These mice recapitulate different aspects of AD pathology, which—at least partially—rely on GSK-3β-induced hyperphosphorylation of tau protein. A similar role for GSK-3α in tau hyperphosphorylation found in AD cannot be precluded, as it has not been extensively analyzed. Abbrev: AHN: Adult hippocampal neurogenesis.

**Figure 2 cells-10-00721-f002:**
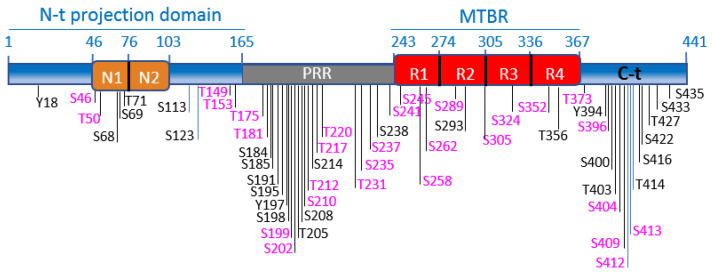
Tau functional domains and phosphorylation sites. Schematic illustration of tau functional domains and localization of the best characterized phosphorylation sites. GSK-3 phosphorylation sites are highlighted in fuchsia. The scheme illustrates the longest tau isoform (tau4R, 441aa). The N-terminal projection domain (aa1-165) contains two inserts, N1 (aa46-75) and N2 (aa76-102). The proline-rich region (PRR) (aa165-242) contains numerous phosphorylation sites, for GSK-3 and other kinases. Tau binds to MTs through the microtubule-binding region (MTBR), which comprises 4 imperfect repeats: R1 (aa243-273), R2 (aa274-304). R3 (aa305-335) and R4 (aa336-367). The C-terminal domain is formed by aa368-441.

**Figure 3 cells-10-00721-f003:**
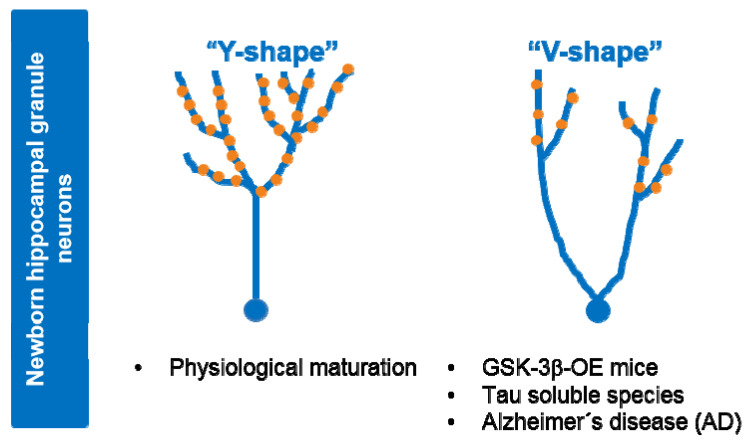
Alterations to dendritic tree morphology and postsynaptic densities in newborn granule neurons upon GSK-3β overexpression or exposure to soluble tau. Under physiological situations (control), maturation of newborn granule hippocampal neurons leads to a dendritic tree with a “Y-shape” that presents a single apical primary dendrite and several distal branches. Upon GSK-3β overexpression (OE) or soluble tau exposure, cells acquire a “V-shape”, with numerous apical dendrites and atrophied distal branching, and show a marked depletion of postsynaptic densities (•PSDs). Neurons from AD patients show similar morphological alterations. At least part of the observed effects of GSK-3β in AHN might be mediated by tau protein.

**Figure 4 cells-10-00721-f004:**
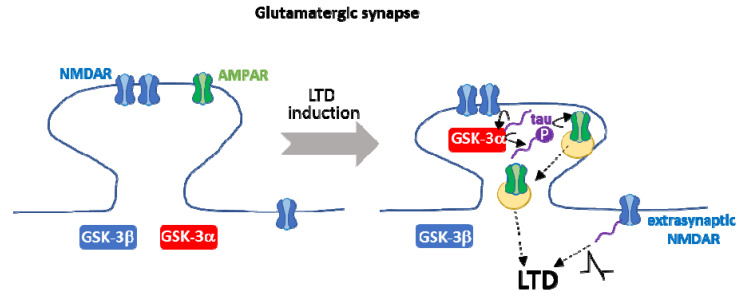
Interplay between GSK-3α and tau during NMDAR-mediated LTD. Tau is required for the transient anchoring of GSK-3α at dendritic spines, a key process during NMDAR-mediated LTD. On the other hand, GSK-3-mediated phosphorylation of tau in Ser 396 is necessary for LTD. Tau contributes further to other crucial aspects of LTD, such as the internalization of AMPAR and NMDAR extrasynaptic currents. It is still controversial whether tau is upstream and/or downstream of GSK-3α in LTD.

## Data Availability

Not applicable.

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
