# Peer review of "GSK-3 and Tau: A Key Duet in Alzheimer’s Disease"

_cells, 2021, doi:10.3390/cells10040721_

Round 1
Reviewer 1 Report
This is a timely and comprehensive review from a leading lab in this field of the roles of GSK-3 with respect to neurodegenerative disease, with a particular focus on Tau hyperphosphorylation. There are some comments and changes suggested below.
- Line 28. While one might, at a stretch, classify yeast and other single cell eukaryotes as invertebrates, plants also show remarkable conservation.
- Line 41. D. discoideum
- Line 48 (and see below) symbol missing? beta?
- Line 63. GSK-3alpha knockout males are infertile. https://pubmed.ncbi.nlm.nih.gov/29385396/
- Line 74, something is missing before "peptide", Abeta? See also line 77 and 78. Seems to be an issue displaying some greek symbols.
- Line 108. While the best evidence for a role for GSK-3 in tau phosphorylation emerges from transgenic expression of GSK-3beta, this does not preclude an equal role for the other isoform, as it largely was not tested.
- Line 133, beta is missing?
- In the paragraph beginning at line 128 there should be mention of other Tau phosphorylating protein kinases. It would also be good to include a diagram of Tau and location of the phosphorylation sites and the identified kinases.
- There are substantive differences between mouse and human Tau and this should be mentioned with respect to Tau and NFTs.
- Line 228, a symbol is missing.
- Line 295 and elsewhere. It is likely the reader will wonder how, with so many processes and proteins regulated by GSK-3 that ascribing certain behaviours and functions to the kinase/substrates is achieved. It might be good to insert a note concerning indirect and direct effects and difficulties in drawing molecular conclusions or rationale for particular brain functions.
- Line 30 - missing symbol.
- There is no mention of collapsin response mediator proteins and regulations of axonal specification. Seems to be applicable in a couple of sections.
- The article jumps to an abrupt and quite short end. This last section could be expanded to discuss the types of therapeutic approaches that might be useful (altering splicing?) and the challenges to this given the unknown implications of messing with Tau. As the authors have made important contributions in this area, why do they think GSK-3 targeting hasn't shown therapeutic benefit, at least in tauopathy? Mention could also be made of the anti-phospho-Tau antibody therapeutics. This is an opportunity to speculate and to place the preceding review into context.
Author Response
Reviewer 1:
Comments and Suggestions for Authors
This is a timely and comprehensive review from a leading lab in this field of the roles of GSK-3 with respect to neurodegenerative disease, with a particular focus on Tau hyperphosphorylation. There are some comments and changes suggested below.
- We thank the reviewer for the positive comments. We changed the text according to the reviewer´s suggestions in the revised version of the manuscript. We respond, point-by-point, to the raised comments below.
Line 28. While one might, at a stretch, classify yeast and other single cell eukaryotes as invertebrates, plants also show remarkable conservation.
- We have changed the sentence to “GSK-3 is evolutionary conserved in all eucaryotes examined to date”, to avoid misleading interpretations and including yeasts, other single cell eucaryotes and plants altogether.
Line 41. D. discoideum
- The typo has been corrected in the text.
Line 48 (and see below) symbol missing? beta?
- Indeed, the b symbol was missing from “amyloid b (Ab) peptide”. It has been amended accordingly in the text.
Line 63. GSK-3alpha knockout males are infertile. https://pubmed.ncbi.nlm.nih.gov/29385396/
- This has been included in the text, together with the reference. Thanks for the reminder.
Line 74, something is missing before "peptide", Abeta? See also line 77 and 78. Seems to be an issue displaying some greek symbols.
- Indeed, the Greek b symbol was missing from Ab in different parts of the text, due to unwanted changes in the letter font. Thank you for bringing this issue to our attention.
Line 108. While the best evidence for a role for GSK-3 in tau phosphorylation emerges from transgenic expression of GSK-3beta, this does not preclude an equal role for the other isoform, as it largely was not tested.
- We agree with the reviewer and thank him/her for the comment. We have included a sentence referred to this issue, at the end of the legend of Figure 1, as follows: “A similar role for GSK-3a in tau hyperphosphorylation found in AD cannot be precluded, as it has not been extensively analyzed”.
Line 133, beta is missing?
- The beta symbol was included after GSK-3 in line 133.
In the paragraph beginning at line 128 there should be mention of other Tau phosphorylating protein kinases. It would also be good to include a diagram of Tau and location of the phosphorylation sites and the identified kinases.
- We have mentioned that tau is phosphorylated by other kinases and named some of them. A scheme illustrating tau domains and the localization of the most relevant phosphorylation sites in tau was also included in the review, highlighting the sites phosphorylated by GSK-3 (since this is the topic of the review). This is the new Figure 2. The rest of the figures have been renumbered (as have the lines of the text). Since we included this scheme on tau structure, to put it into context we had to include also some general information about tau functional domains, splicing, and isoforms. For that reason, we included a new general section on tau protein.
There are substantive differences between mouse and human Tau and this should be mentioned with respect to Tau and NFTs.
- A paragraph on the differences between mouse and human tau, mentioning different developmental regulation of tau isoforms and at the N-terminal domain, has been included in the text (new section on tau) and we referred to the propensity of human tau to be involved in pathological processes such as NFT formation.
Line 228, a symbol is missing.
- The icon was added where needed.
Line 295 and elsewhere. It is likely the reader will wonder how, with so many processes and proteins regulated by GSK-3 that ascribing certain behaviours and functions to the kinase/substrates is achieved. It might be good to insert a note concerning indirect and direct effects and difficulties in drawing molecular conclusions or rationale for particular brain functions.
- A short note on this issue has been included in the first paragraph of the Conclusions section.
Line 30 - missing symbol.
- The beta symbol was added after GSK-3.
There is no mention of collapsin response mediator proteins and regulations of axonal specification. Seems to be applicable in a couple of sections.
- Our review is focused on the interplay between GSK-3 and tau in the context of biological processes that are dysregulated in AD. Therefore, although collapsin response mediator proteins (CRMPs) are important substrates of GSK-3 phosphorylation during axon specification and in AD, this topic would be out of the focus of the current manuscript. Furthermore, in a recent article published in the present Special Issue, the relationship between GSK-3 and CRMPs has been treated in some detail (Rippin and Eldar-Finkelmann, Cells 2021, 10(2), 262; https://doi.org/10.3390/cells10020262), and our contribution would be somehow redundant.
The article jumps to an abrupt and quite short end. This last section could be expanded to discuss the types of therapeutic approaches that might be useful (altering splicing?) and the challenges to this given the unknown implications of messing with Tau. As the authors have made important contributions in this area, why do they think GSK-3 targeting hasn't shown therapeutic benefit, at least in tauopathy? Mention could also be made of the anti-phospho-Tau antibody therapeutics. This is an opportunity to speculate and to place the preceding review into context.
- To avoid this jump to an abrupt and short end, we have included an extra entire section on therapeutic approaches that target GSK-3 (GSK-3 inhibitors) or tau phosphorylated in GSK-3 sites. The use of anti-phospho-tau antibodies as a potential therapy for AD has been documented and discussed, as well as the use of tau-directed vaccines (tau-based passive and active immunotherapy). The implications of targeting GSK-3 or tau have also been commented. We have speculated on the importance of the GSK-3 and tau duet as potential therapeutic targets in AD. We have further discussed this issue in a new paragraph in the Conclusions and future perspectives section. A sentence concerning GSK-3 and tau-based therapies has also been added to the Abstract.
Reviewer 2 Report
The manuscript proposed by the authors recapitulates the knowledge on the roles of GSK3 and Tau in Alzheimer's disease. It is well constructed and very clear. I recommend the following minor corrections:
-Please, carefully check the presence of symbols in GSK3 in lines 30, 86, 133, 223, 262, 330 and Fig.2 legend.
-Also, check the presence of symbol in amyloid peptide in the abstract and in lines 74, 77, 78, 358.
-Please, include the significance of AHN abbreviation in the Fig. 1 legend.
-Line 316: It would be better to write "Tau is involved in LTD induced by insulin resistance [151] or glucocorticoids (GCs) [152]."
-Discard spaces in lines 52 and 228 and add a point in line 76.
Author Response
Responses to Reviewer 2:
The manuscript proposed by the authors recapitulates the knowledge on the roles of GSK3 and Tau in Alzheimer's disease. It is well constructed and very clear. I recommend the following minor corrections:
- We thank the reviewer for the positive comments. We include a point-by-point answer below.
-Please, carefully check the presence of symbols in GSK3 in lines 30, 86, 133, 223, 262, 330 and Fig.2 legend.
-Also, check the presence of symbol in amyloid peptide in the abstract and in lines 74, 77, 78, 358.
- We thank the reviewer for bringing our attention to the symbol issue. We have included the a or b Greek symbols where required.
-Please, include the significance of AHN abbreviation in the Fig. 1 legend.
- The definition of AHN was included in the Figure 1 legend, as indicated.
-Line 316: It would be better to write "Tau is involved in LTD induced by insulin resistance [151] or glucocorticoids (GCs) [152]."
- In reference 151, the authors use a protocol of LTD induction by the addition of 1microM insulin to hippocampal slices of either wt or tau-KO mice. In tau-KO mice, the magnitude of LTD was reduced in comparison to wt littermates. Therefore, tau is involved in LTD induced by insulin, as we stated in the text (not in LTD induced by insulin resistance). In that article, the authors also show that tau deletion leads to hippocampal insulin resistance.
-Discard spaces in lines 52 and 228 and add a point in line 76.
- We have discarded the mentioned spaces and included the suggested corrections to the text.